# Study of a Current and Voltage Polarization Sensor Network

**DOI:** 10.3390/s21134528

**Published:** 2021-07-01

**Authors:** Artur de Araujo Silva, Claudio Floridia, Joao Batista Rosolem

**Affiliations:** CPQD—Research and Development Center in Telecommunications, Campinas 13086-902, Brazil; arturs@cpqd.com.br (A.d.A.S.); floridia@cpqd.com.br (C.F.)

**Keywords:** optical sensor network, polarization sensors, current sensors, voltage sensors, DWDM

## Abstract

Sensors based on polarization are suitable for application in power grids due to their excellent characteristics, such as high electrical insulation, non-magnetic saturation, oil-free, no risk of explosive failures, and high bandwidth. Utility companies are incorporating new technologies that are driving the evolution of electrical systems. Thus, it is interesting to evaluate the possibility of using polarization sensors in a network configuration. In this work, we present an experimental study of a current and voltage polarization sensor network applied to a medium voltage distribution grid. The current sensor is based on the Faraday effect, and the voltage sensor uses the Pockels effect. Both sensors use a 90° polarization degree between the two output ports to compensate for the various impairments on the measurements by applying the difference-over-sum. The network uses a DWDM topology centered at the 1550 nm range, and both current and voltage sensors in this work used this spectral band. We evaluated the sensor node in terms of accuracy according to IEC standard 61869-10 and IEC standard 61869-11. Considering that an important application of this sensor network is in the aerial cable of medium voltage networks, sensor node accuracy was also estimated in the presence of cable vibration. The calculated power budget of the proposed network indicates that reaching ten nodes of current and voltage sensors in a 10 km optical link is possible, which is enough for a medium urban voltage distribution network.

## 1. Introduction

Optical fiber sensors have attracted attention recently due to their many advantages, such as small, lightweight, high sensitivity, immune to electromagnetic interferences, multi-parameter capacity, and ease of connection to a network. Optical sensor networks lead to the possibility of reducing the cost of the total sensing system since the network sensors will share the same interrogation unit [1].

A relevant aspect of optical fiber sensor networks is the topology. A fiber optic sensor network usually includes an array of point optical sensors, a system that allows the multiplexing of sensor signals. The sensor signal can be multiplexed by time-division multiplexing (TDM), wavelength division multiplexing (WDM), frequency division multiplexing (FDM), and space division multiplexing (SDM) [1,2,3]. Another issue in fiber optic sensor networks regards the protection schemes [4,5] since, in many situations, the replacement of a damaged sensor is not easy.

Although there are studies on optical sensor networks for the most technologically significant types of sensors, such as interferometric [3], Fabry-Perot [6], fiber Bragg grating [7], and fiber bending [8], there are no studies for polarization optical sensors.

In this context, polarization optical current and voltage sensors using Faraday and Pockels effects have been studied since the 1990s as they can replace conventional current transformer (CT) and voltage transformer (VT) sensors [9]. The working principle of optical CTs is to measure the magnetic field generated by an electric current using light modulation and demodulation by the Faraday effect. Optical VTs are used to measure voltage using the Pockels effect. If linearly polarized light is injected into the sensor when a high voltage is applied to this element, the light is divided into two linearly polarized beams in x and y directions at different speeds. This difference in propagation velocity causes the beams to cross at right angles, thus becoming an elliptically polarized beam with a phase difference between them at the sensor output. This phase difference is proportional to the voltage applied to the sensor element [9].

Sensors based on polarization are suitable for power grid applications due to their excellent characteristics, such as high electrical insulation, non-magnetic saturation, oil-free, no risk of explosive failures, and wide bandwidth [10,11]. Utility companies are incorporating new technologies that are driving the evolution of electrical systems. Thus, it is interesting to evaluate the possibility of using polarization sensors in a network configuration.

In this work, we present an experimental study of a current and voltage polarization sensor network designed to be used in a medium voltage distribution grid. To the best of our knowledge, this is the first demonstration of an optical network using polarization sensors. The use of DWDM systems with add/drop is typical in the telecom sector, but the literature does not report the use of DWDM-sensing networks for voltage and current polarization sensors. One factor that can contribute to this fact is that most current and voltage sensors used in the literature do not use telecom wavelengths of 1550 nm. For example, many current sensors use 630 nm as the wavelength in TGG [12]. The choice of RIG and LiNbO3 materials allows these sensors to be applied to network use in the C-Band wavelength telecom range. The current sensor is based on the Faraday effect, and the voltage sensor uses the Pockels effect. Both sensors use 90° polarization degrees between the two output ports to compensate for the various impairments on the measurements by applying the difference-over-sum. The network uses a DWDM topology centered at the 1550 nm range, and both current and voltage sensors in this work used this spectral band. We evaluated the sensor node in terms of accuracy according to IEC standard 61869-10 [13] and IEC standard 61869-11 [14]. We also evaluated the sensor node in terms of vibration since one of the significant applications of this sensor network is in aerial cables of medium voltage networks. The paper is organized as follows: Section 2 presents the optical sensor network topology and details of the optical source and the optical sensor node. Section 3 shows the experimental characterization of a complete current and voltage sensor node. Section 4 presents the results, discussion, and a calculation of the optical power budget of the network using the parameters of commercial devices. 

## 2. Dual Output Voltage and Current Sensor Scheme

The proposed sensors use the one input, two optical outputs architecture, well-studied in optical sensors based on the Faraday effect for current measurement and the Pockels effect for voltage measurement [15,16]. These architecture are used to compensate for various impairments of the measurements by applying the difference-over-sum in the acquired waveforms. This method was proposed in 1989 [15] and has been extensively used in these sensors. In this technique, the laser beam is divided into two components with mutually orthogonal polarization planes by the crystals inside the sensor. The light power of both beams depends on the source intensity in the same way. When the difference-over-sum method is applied, this dependence on source intensity is removed. In [15,16,17], the method is completely described. The effects of optical power fluctuations [16], residual polarization state [17], and multiple birefringences [15] are also compensated. The physical reasons for these impairments are light source oscillations and losses in the optical link; in addition, the residual polarization of the ASE source has a polarization state that changes with the movement of the optical fiber.

In this work, the current sensor used a common rare-earth iron garnet (RIG) employed in the assembly of fiber optical components such as Faraday mirrors, rotators, and circulators. RIG has an estimated equivalent Verdet constant of ~22,300 rad/(T m) @ 1550 nm, 166 times bigger than TGG, although its Faraday rotation is quite different from the magneto-optical effect that is proportional to the external magnetic field [17,18,19]. The voltage sensor used a LiNbO_3_ crystal [17]. 

The employed current sensor and voltage sensors were embedded in a single package, designed to be employed in aerial cables of medium voltage distribution networks. Thus, the proposed voltage and current optical sensor uses two input fibers and four output fibers to determine the current and voltage of the conductor where it is introduced, as shown in Figure 1. 

Figure 1a shows both the internal structure of the sensors and their pictures. The VS is composed of a fiber optical collimator, one polarizer oriented @ 45°, a quarter-wave plate with a fast axis at the vertical direction, a z-cut LiNbO3 crystal, a polarization beam splitter oriented @ 45°, and a dual fiber optical collimator. The structure of the CS is similar. It is composed of a fiber optical collimator, a polarizer oriented @ 0°, a magneto-optical material, a polarization beam splitter oriented @ 45°, and a dual fiber optical collimator. Figure 1b shows the sensor head. The input and output optical fibers are placed inside an insulator and arrive at a box where the VS is placed in a specially designed electrode. The CS is placed laterally in a movable bipartite core that concentrates the magnetic field; the optical fibers of the CS sensors are also accommodated in the optical fiber tray together with the sensor itself. The bipartite core is used to enable the placement of the whole structure in the conductor of the distribution line.

## 3. Sensor Network Topology

The sensor network used in this work uses dense wavelength division multiplexing (DWDM) to attend to the sensor nodes, as shown in Figure 2. The main elements of the network are the amplified spontaneous emission (ASE) light source, an interleaver to slice the spectrum, the sensor nodes, the demultiplexers, and the optical receivers, as shown in the inset of Figure 2. The interleaver device has two outputs called “Odd” and “Even”, where the λ_1_, λ_3_, λ_5_, and λ_2_, λ_4_, λ_6_ wavelengths are present, respectively. The set of wavelengths are then forwarded to the Add/Drop devices placed in the sensor node, which separates a specific wavelength and directs it to the input of a current or voltage sensor. Both sensor outputs are returned to the network via another Add/Drop device. Finally, all wavelengths are demultiplexed and demodulated in pairs, and the signal is sent to optical receivers (RXs) in the processing unit.

To better understand the sensor node, we refer to the inset of Figure 2. λ_1_ from the odd wavelength in the upper fiber of the figure is drooped by the A1 add/drop and is directed to the VS sensor. The first output of the VS is sent to the B1 add/drop that re-adds this wavelength to the fiber. This signal passes unperturbed (unless an insertion loss happens) through the C1 add/drop. The second output of the VS sensor is sent to the C2 add/drop in the lower branch of the node and re-added to the second fiber. Similarly, the λ_2_ at the lower fiber optics of the inset is dropped and redirected to the CS by the A2 add/drop; the first CS output is directed to the upper fiber and added by the C1 add/drop. The second output of the CS is added again to the lower fiber by the B2 add/drop. This signal passes unperturbed (unless an insertion loss happens) in the C2 add/drop and travels through the lower fiber to the other nodes. The pair after this node, λ_1_ and λ_2_, will not be affected by subsequent nodes because they add and drop other wavelengths.

In a standard application, the processing unit of each sensor is installed inside an enclosure at the bottom of the pole, and the data are transmitted to a control room using general packet radio services (GPRSs). In our proposed application, just one processing unit is installed in a control room inside a substation. The signals can be transmitted from the sensors to this control room using optical fibers of the distribution network. In this way, this proposed network can save many processing units installed along with the distribution network and increase the robustness of the measurement system. Besides that, the cost of add/drops is relatively low compared to the sensor elements themselves, widely due to mass production for the telecom sector.

Figure 3a shows the signal spectrum of the ASE light source at the interleaver input, the “Even” and “Odd” ports of the interleaver device, and the signals from the current and voltage sensor inputs, separated by Add/Drop. In addition, Figure 3b,c show the spectrum of the input and the signals of the voltage (VS) and current (CS) sensors at RX after they pass through DEMUX, respectively. To obtain the spectrum, we used an optical ASE source and an optical spectrum analyzer (OSA), Anritsu MS9740A.

## 4. Light Source Selection

As a first step to developing and testing the proposed DWDM sensor network, we carried out an extensive evaluation of possible ASE sources to be used in the sensor system. In these analyses of light sources, we considered two main parameters: the power of the light source and its degree of polarization (DoP). As seen in [20], decreasing the power of the light source can gradually distort the sensor’s response. This distortion can happen due to the decrease of the signal-to-noise ratio (SNR) of the light source. Furthermore, as shown in [18], the higher the DoP of the light source, the greater the variation of the state of polarization (SoP), resulting in a higher variation in the responses of the two optical outputs.

The ASE source needs to have the highest power available and be stable in its operation to avoid unwanted power fluctuations and to attend to as many nodes as possible. Moreover, the ASE source must have the lowest possible degree of polarization (DoP). Even if the used sensor design permits the mitigation of output fluctuations due to the residual state of polarization of the input light, as demonstrated by the authors in [18], the use of completely depolarized light would be ideal in such a system. 

To generate the necessary current, we used a variable transformer (varivolt), a power resistor, and a coil of 30 turns. To measure the applied current, we used a clamp meter (Minipa CA-1000) connected to an oscilloscope (Keysight InfiniiVision MSOX2024A). For voltage measurement, a high voltage source (Exactus EAT33 1452-B) was used, and a high voltage probe (Tektronix P6015A) was used to measure the applied voltage.

To take the measurements, we evaluated several light sources, shown in Table 1. The “Compact ASE Source” is an EDFA, model NOAPF201X00001X from Lighwaves2020, with 8 dBm total emitted power. The light source “High Power EDFA” is an EDFA of 20 dBm output power. However, to be activated, it needs minimum input power due to the automatic shutdown protection criteria. If there is no input, the equipment emits a low-power mode of 4 dBm (Source B, Table 1). The C source type is a high-power EDFA with 1% of its power fed back to its input using an optical coupler. These three operation types of high-power EDFAs are named Operation Types 1, 2, and 3 in Table 1. The D source type is a high-power EDFA using optical input power provided by Lightwaves2020 but attenuated by 40 dB. Source E is a SLED source, which is known to have a DOP of 27.6% [18].

To evaluate system performance with each optical source of Table 1, we tested the current sensor response by varying the applied current from 10 to 120 A. The rated current was 100 A for this experimental test. We analyzed current sensor accuracy based on IEC standard 61869-10 [13], which establishes the percentage ratio error class limits as a function of the rated current from 5% to 120%. The sensor response in the above range showed the expected nonlinear behavior, which can be fitted by an arcsine function.

We calculate the least square method (LSM) of the ratio error for each applied rated current to determine the best ASE source. This parameter gives the deviation from the zero-ratio error of the ASE source in the entire range of rated currents, which is the overall accuracy of the system. Table 1 shows these values, together with the source name and power. Other criteria that were used to determine the best ASE source are the sum of the error bars shown in Figure 4 or $\sum \sigma_{i}$. This criterion indicates how much each measurement is dispersed from the other, an indicator of overall precision, the best source being the one with better overall accuracy and overall precision. The best choices from Table 1 are Sources A and E. Source E is the SLED source, which has a DOP of 27.6%. Although Source E was the best in that criterion, we discarded this source because the individual optical outputs had a substantial variation in power levels when submitted to external perturbation (SOP variation). Such variation may cause the saturation of optical receivers. Thus, we chose Source A for our application.

## 5. Results and Discussion

After defining the best ASE source for the sensor, we used this source to measure the sensors with a DWMD network, as in Figure 1, and without a DWMD network, as proposed by this study in Figure 2. These tests, with and without a DWDM network, were carried out to compare the sensors’ performance in these situations. A considerable change between the two setups is the sensors’ optical input since the bandwidth of sensor input in the DWDM network is around 0.5 nm. Without the DWDM network, the bandwidth is approximately 60 nm. Additionally, it is relevant to verify if the add/drop devices introduce any effect in the sensors due to their polarization-dependent loss (PDL). Once the sensors were applied in aerial conductors, we performed vibration tests to consider this operating condition. Hence, they were subjected to environmental disturbances such as winds and rains. Although vibration tests are standardized for high current and voltage sensors, such as in [21], these are not relevant to this application because these standards refer to substation vibration when switching gears or vibrations due to earthquakes. Some papers have addressed the vibration effects and compensation in optical fiber current sensors. In [22,23], a frequency signal of 55 and 200 Hz was applied to an optical fiber current sensor, respectively. In [24], the study of vibration was performed in the range of 20 to 500 Hz. However, all these sensors were intended to be applied in substations in high power plants. In our application, we used a low frequency (less than 60 Hz) vibration on the optical cables as the main issue was the influence of wind vibration in these cables. Appendix A shows the effect of cable perturbation on the two optical output signals (upper and lower traces of the oscilloscope) of the current sensor. Compensation is achieved (central trace of the oscilloscope) by adding these two optical output signals. The applied current was increased during the video, and the 60 Hz current signal was observed to increase accordingly.

The results were obtained by measuring the sensor response for the current sensor with and without vibrational perturbation. Furthermore, we measured the sensors’ response with and without a DWMD-employed network. The current was increased from 10 to 240 A. The rated current for this sensor is 200 A. We employed IEC standard 61869-10 [13], which establishes the percentage ratio error class limits. In the same way, results were obtained for the voltage sensor. In this case, we employed IEC standard 61869-11 [14], which establishes the percentage ratio error class limits for voltage sensors. This limit ranges from 80% to 120% of the rated voltage. The rated voltage was 13.8 kV root-mean-square (RMS) phase-to-phase or 7.97 kV (RMS) phase-to-ground.

### 5.1. Current Sensor

We perform the measurements of the current sensor by increasing the applied current from 10 to 240 A. The reference signal was registered as well as the outputs of the sensors. The two output optical signals in the receivers were summed and amplified by proper detection circuitry, giving rise to the current sensor measurement. The rated current for this sensor was 200 A. Thus, a variation from 5% to 120% of the rated current was performed. 

#### 5.1.1. Without DWDM

By applying the current in the prototype in the traditional form of operation (without a DWDM network), we obtained the results shown in Figure 5. This plot is obtained by measuring the optical current sensor root-mean-square (RMS) response at each applied current, considering ten waveforms of 200 ms of a duration corresponding to 12 cycles at 60 Hz AC. As can be seen, the data is within the 1.0 class limit of IEC standard 61869-10 [13], mainly in terms of accuracy (ratio error). On the other hand, the sensor presents high precision as its error bars are of the same order of magnitude as the dots on the plot.

An issue regarding medium voltage optical sensors is related to the action of wind on the cable or external perturbation that moves the sensor cable during operation. Usually, these sensors are installed in the conductors of the power distribution grid, and its optical cable is free to move. 

Thus, the red points in Figure 5 were obtained by applying a perturbation in the input optical fiber of the sensor, corresponding to vibration or wind action on the cable. This results in the variation of the residual state of polarization, as reported in [18]. The individual optical outputs of the sensor fluctuate, as illustrated at the top of Figure 6, but it is compensated by the summation of these traces, as shown at the bottom of Figure 6. As can be seen, the compensation is quite good and can maintain the reported class limit of the sensor within the 1.0 class. This perturbation was applied close to the values of 5, 25, 50, 75, 100, and 125 percent of the rated current.

#### 5.1.2. With DWDM

For the sensor inserted in the DWDM network, we repeated the tests. We observe that without any perturbation, the current sensor practically maintains its response within the 1.0 class limits; only for the 5% rated current, there is a slight limit deviation. This result is shown in Figure 7. However, with the perturbation applied to the optical fiber, the 1.0 class limits are exceeded mainly for small currents (below 30% of the rated current). It is also significant to observe that the optical power level in the sensor output was kept constant at −23 dBm, the same level as the no DWMD test, by employing optical attenuators at the input.

### 5.2. Voltage Sensor

We performed the measurements of the voltage sensor by increasing the applied voltage from 5.57 to 9.56 kV. The reference signal was registered as well as the outputs of the sensor. The two output optical signals in the receivers were subtracted and amplified by the circuitry, giving rise to the voltage sensor measurement. The rated voltage for this sensor was 7.97 kV. Thus, a variation from 70% to 120% of the rated voltage was performed; IEC standard 61869-11 [14] requires that sensors are within the class limit, from 80% to 120%, of the rated voltage.

#### 5.2.1. Without DWDM

Firstly, we carried out measurements of the voltage sensor without the presence of the DWDM sensor network. This experiment was done by measuring a 200 ms signal 10 times for each applied voltage. This applied voltage was varied from 70% to 120% of the nominal voltage.

The procedure to obtain the error for each applied voltage was the same as the current sensor described in the previous section. Figure 8 shows the results of the measurements. It is possible to notice that the sensor is classified as class 0.5%. However, in the presence of disturbance, this response undergoes a considerable change, leaving class 1%. This result can be explained due to the lower signal-to-noise ratio (SNR) of the voltage sensor compared to the current sensor.

Figure 9 shows the waveforms, in the presence of disturbance, of the two output channels of the sensor as well as the result of the subtraction and amplification that gives rise to the sensor signal.

#### 5.2.2. With DWDM

Finally, we added the components that make up the DWDM sensor network to the voltage sensor. Therefore, it was possible to repeat the measures mentioned above. Figure 10 shows the result of these measurements of the voltage sensor in the presence of the DWDM network. It is noted here that the influence of the disturbance was less than in the case without a DWDM network. Thus, the sensor is characterized in the 1% class in both cases. The output power of the two voltage sensor channels was kept the same as in the case without a DWDM network, applying attenuators to the sensor input.

Figure 11 shows the waveforms, in the presence of perturbation in the optical input cable, of the two output channels of the sensor as well as the result of the subtraction and amplification that gives rise to the sensor signal with the DWDM network.

### 5.3. Sensor Network Capacity

This section presents the calculation, results, and discussion of the optical power budget of the network using the parameters of commercial devices. The use of sensors in a network to monitoring large-scale power distribution grids was presented and discussed in [25]. In this paper, many sensors were installed at the branching points of electrical conductors. This proposed scheme allowed the estimation of the grid’s topology and monitored the integrity of the network. The sensors were separated into clusters, composed of the sensors installed on a single-line branched conductor; they used ten sensors. The average distance of a feeder in a medium voltage distribution network is around 11.53 km for the suburban network and 4.46 km in urban networks [26]; thus, a calculation is possible.

The study to expand our proposed sensor node to form a network with several nodes should follow these guidelines. To verify the possibility of this expansion and to estimate the number of sensor nodes that a network can support, power and loss data from commercial devices were considered in this study. Equation (1) was used to estimate the optical power budget of the sensor network.
(1)$$P_{\mathrm{ASE}}-\left[A_{\mathrm{Slice}}+{\;\mathrm{IL}}_{\mathrm{Int}}+{\mathrm{IL}}_{\mathrm{Ad}}\times N+{\mathrm{IL}}_{\mathrm{Sensor}}+{\mathrm{IL}}_{\mathrm{Demux}}+L\times \alpha_{\mathrm{Fiber}}\right]\geq -23\;\left[\mathrm{dBm}\right]$$
where P_ASE_ is the total power of the ASE source in dB within its spectral band B_ASE_, A_Slice_ is the optical attenuation caused by the slicing of the ASE power by the interleaver, assuming the spectral flat ASE power A_Slice_ is given by A_Slice_ = 10.log_10_ (B_ASE_/B_ch_), where B_ch_ is the spectral band of the interleaver channel, IL_Int_ is the intrinsic insertion loss of the interleaver in dB, IL_Ad_ is the add/drop insertion loss in dB, N is the number of nodes of the network, IL_Sensor_ is the insertion loss of a sensor (current or voltage) in dB, IL_Demux_ is the insertion loss of the DWDM demultiplexer in dB, L is the total length of the fiber link in km, and α_Fiber_ is optical fiber attenuation in dB/km. We consider −23 dBm as the minimum optical power in the optical receivers to guarantee the accuracy levels of the sensors, according to the measurements presented before. In this approach, we disregarded the insertion loss of the optical connectors and splices.

To compute the maximum node quantity in the network, we chose the parameters of (2) using data from commercial devices. The launch power of the $P_{\mathrm{ASE}}$ source was 23 dBm, with B_ASE_ = 30 nm. The loss of spectrum slicing, performed by the interleaver $A_{\mathrm{Slice}},$ was 17.78 dB for a 30 nm device, with B_ch_ = 0.8 nm (100) GHz bandwidth for each channel; additionally, a 2 dB insertion loss, $IL_{\mathrm{Int}}$, was considered. The insertion loss of the sensor was ${\mathrm{IL}}_{\mathrm{Sensor}}=4\;\mathrm{dB}$. The light will pass through three Add/Drop devices in each sensor node, thus the insertion loss of the Add/Drop, ${\mathrm{IL}}_{\mathrm{Ad}}=0.5\mathrm{dB}\times 3$, will be multiplied by the number of nodes, N. A loss caused by 10 km of optical fiber link, $L_{\mathrm{Fiber}}=2.5\;\mathrm{dB}$ and the attenuation of the demultiplexer $A_{\mathrm{DEMUX}}=4\;\mathrm{dB}$, were also considered. If N = 10, these losses will result in optical receiver power of −22.28, which is greater than −23 dBm. Then, the calculated power budget of the network indicates that it is possible to reach 10 nodes of current and voltage sensors in a 10 km optical link. This number of sensors is in accordance with the design guidelines applied to medium urban voltage distribution networks [25,26]. Figure 12 shows a plot of the number of nodes versus the link length (L) using (1) and the data cited above for three levels of ASE source optical power P_ASE_.

## 6. Conclusions

Considering that utility companies are incorporating new technologies to drive the evolution of electrical systems, it is interesting to evaluate the possibility of using polarization sensors in a network configuration. In this work, we have proposed a current and voltage sensor network applied to a medium voltage distribution grid. The proposed sensor network was tested, and the results were compared with the sensors working conventionally, that is, alone or without a network.

The current sensor was evaluated for a range from 5% to 120% of the rated current of 200 A. For the case of the sensor without a DWDM network, it was classified as a 1% class sensor. This classification was maintained even with the presence of vibration in its optical cable.

In the DWDM configuration, the 5% rated current point exceeded the 1% class line threshold. When a disturbance was applied to this sensor, the 25% rated current point also exceeded the 1% class limit.

The voltage sensor was tested for a range of 70% to 120% of the rated voltage of 7.97 kV. For the case without a DWDM network, it was noted that the sensor had a ratio error curve that did not exceed the class limit of 0.5%, although it had an error bar above this limit. When a disturbance was applied to the sensor’s optical cable, it was noted that it exceeded the class limit of 1%. With the presence of the DWDM network, the sensor showed similar behavior to cases with and without disturbance, staying within the 1% class.

Finally, the calculated power budget of the network indicates that it is possible to reach 10 nodes of current and voltage sensors in a 10 km optical link, which is in accordance with the design guidelines applied for medium urban voltage distribution networks.

## Figures and Tables

**Figure 1 sensors-21-04528-f001:**
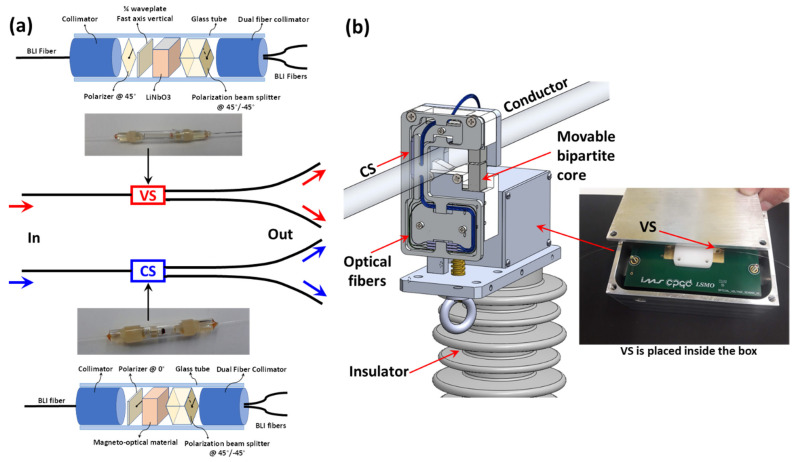
(**a**) Optical setup of the voltage (VS) and current (CS) optical sensors; (**b**) sensor head showing details of the placement of the VS and CS in the packaging.

**Figure 2 sensors-21-04528-f002:**
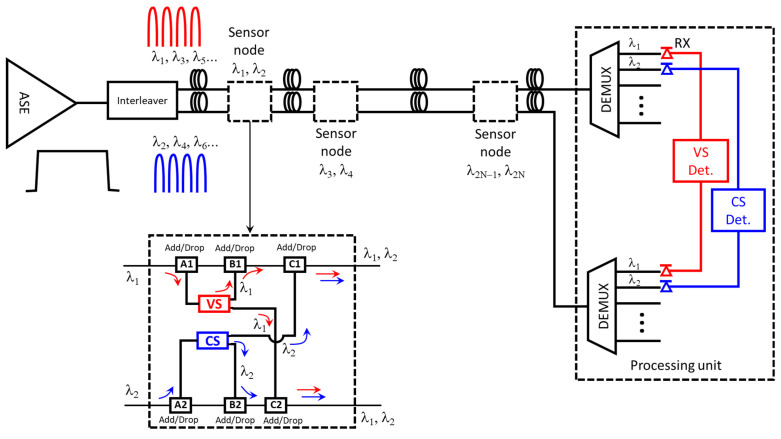
The general design of the current and voltage sensor network. An ASE light source, an interleaver, a current sensor using the Faraday effect, a voltage sensor using the Pockels effect, add/drop to separate the spectrum, and DEMUX for separate demodulation of the current and voltage signals are used.

**Figure 3 sensors-21-04528-f003:**
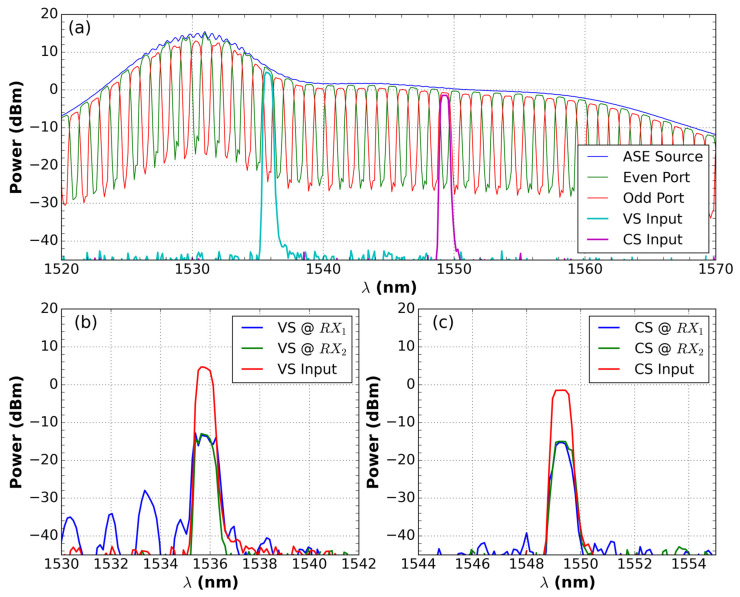
(**a**) Signal spectrum of the ASE light source at the interleaver input, the “Even” and “Odd” ports of the interleaver device, and the signals from the current and voltage sensor inputs, separated by Add/Drop. In addition, (**b**) the spectrum of the input of the voltage sensor (VS) and signals of the VS @ RX after passing through DEMUX; (**c**) the spectrum of the input of the current sensor (CS) and signals of the CS @ RX after passing through DEMUX.

**Figure 4 sensors-21-04528-f004:**
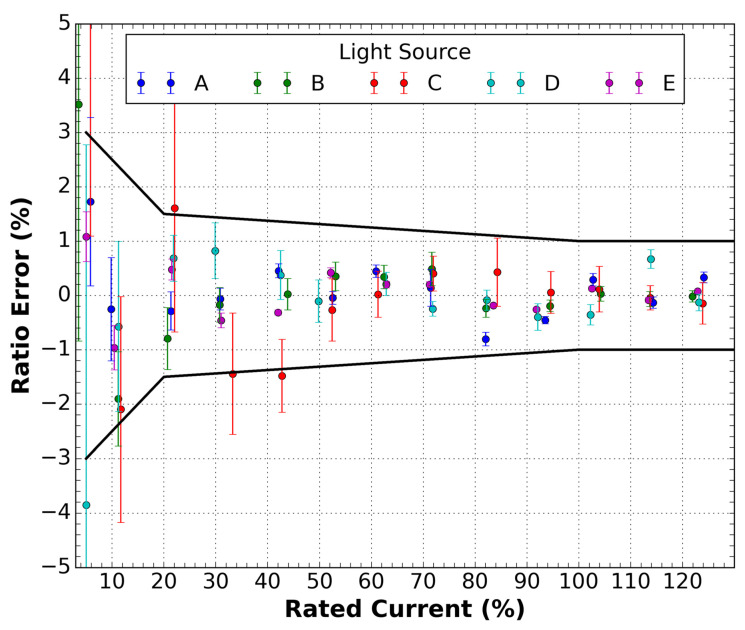
Ratio error curve as a function of the rated current for various evaluated light sources.

**Figure 5 sensors-21-04528-f005:**
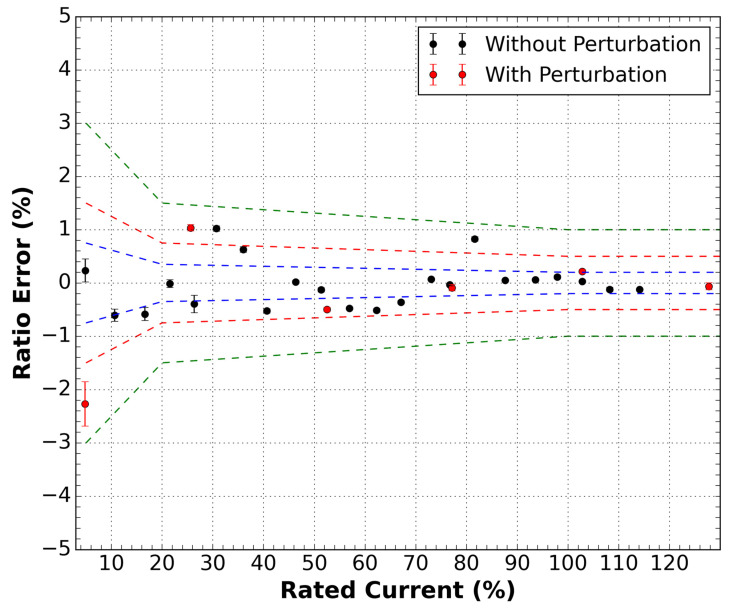
Ratio error measurements as a function of the rated current for the sensor outside the DWDM network. Two conditions were measured, with and without disturbance in the optical cable of the sensor. Green dashed lines are the 1.0 class limits, red dashed lines are the 0.5 class limits, and the blue dashed lines are the 0.2 class limits.

**Figure 6 sensors-21-04528-f006:**
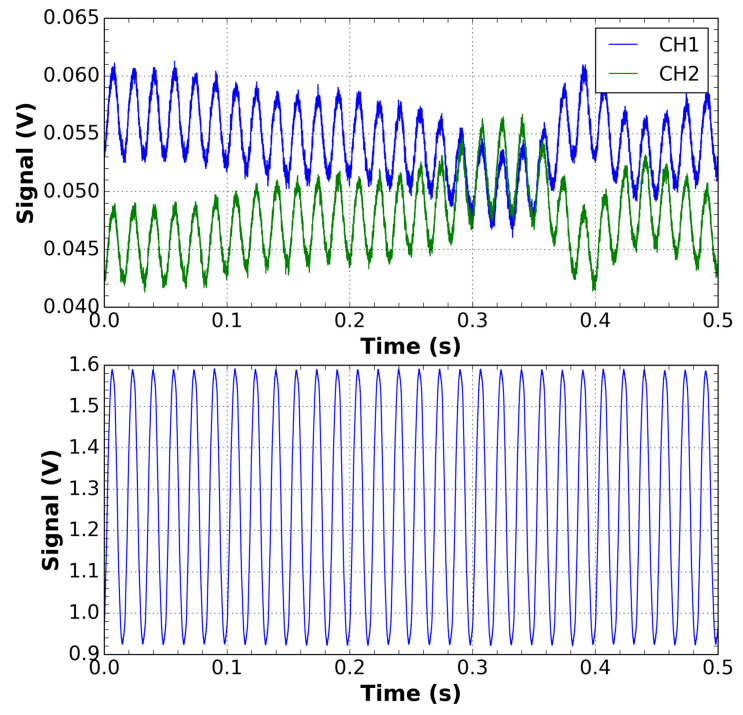
Waveforms of the CH1 and CH2 channels of the current sensor output in the presence of a disturbance with an applied current of 240 A.

**Figure 7 sensors-21-04528-f007:**
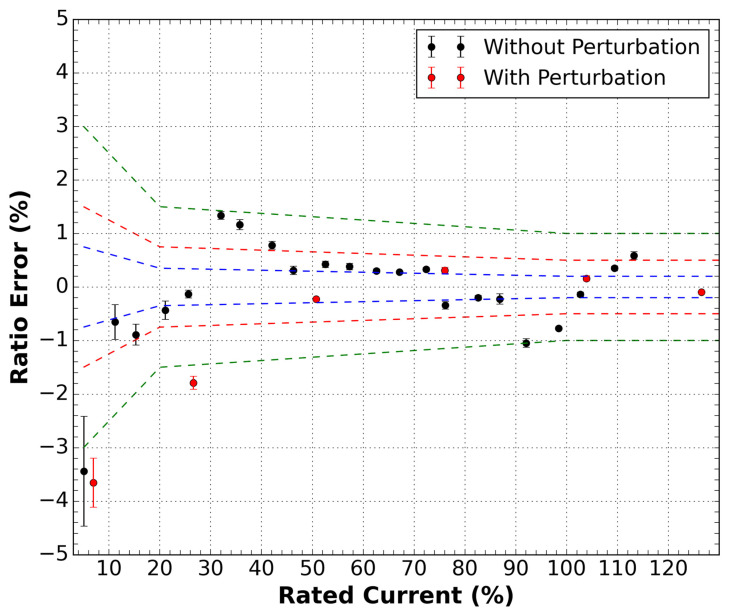
Ratio error measurements as a function of the rated current for the sensor inside the DWDM network. Two conditions were measured, with and without disturbance in the optical cable of the sensor. Green dashed lines are the 1.0 class limits, red dashed lines are the 0.5 class limits, and the blue dashed lines are the 0.2 class limits.

**Figure 8 sensors-21-04528-f008:**
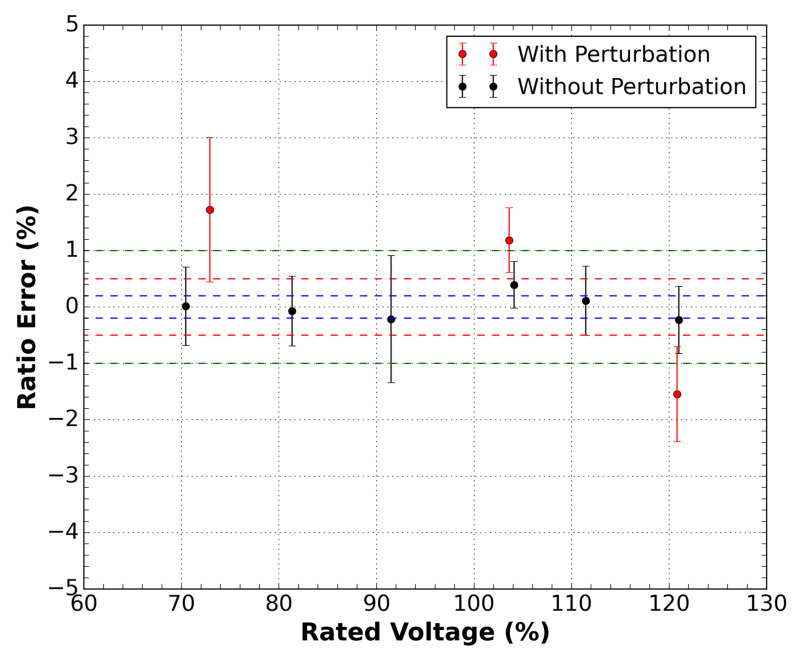
Ratio error measurements as a function of the rated voltage for the sensor outside the DWDM network for the conditions with and without disturbance in the optical cable of the sensor. Green dashed lines are the 1.0 class limits, red dashed lines are the 0.5 class limits, and the blue dashed lines are the 0.2 class limits.

**Figure 9 sensors-21-04528-f009:**
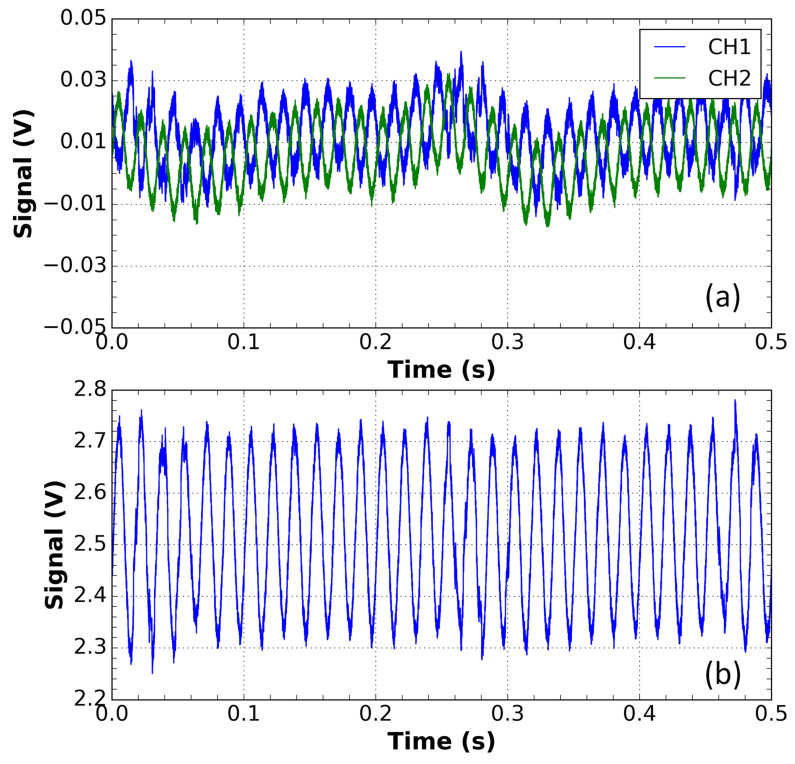
Measurements for the sensor outside the DWDM network. (**a**) Waveforms of the CH1 and CH2 channels of the voltage sensor output in the presence of a disturbance with an applied voltage of 9.56 kV. These signals were subtracted to compensate for the amplitude variation; (**b**) the result of this subtraction.

**Figure 10 sensors-21-04528-f010:**
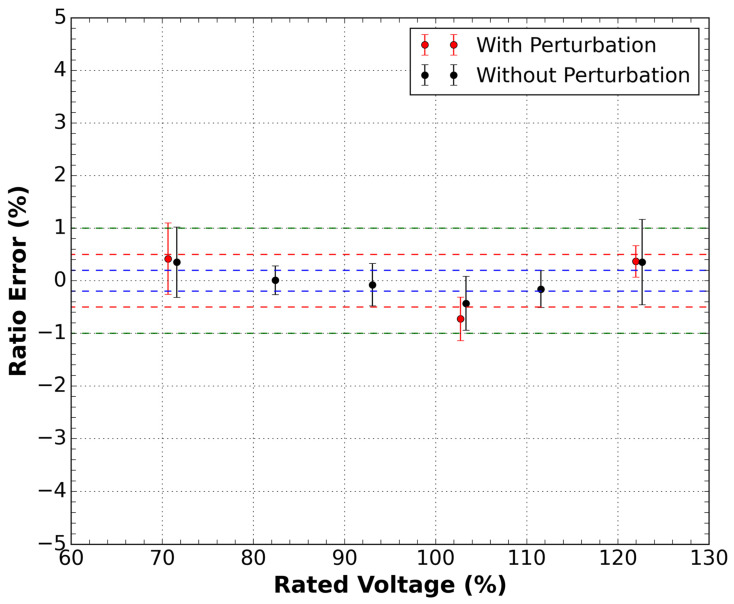
Ratio error measurements as a function of the rated voltage for the sensor inside the DWDM network for the conditions with and without disturbance in the optical cable of the sensor. Green dashed lines are the 1.0 class limits, red dashed lines are the 0.5 class limits, and the blue dashed lines are the 0.2 class limits.

**Figure 11 sensors-21-04528-f011:**
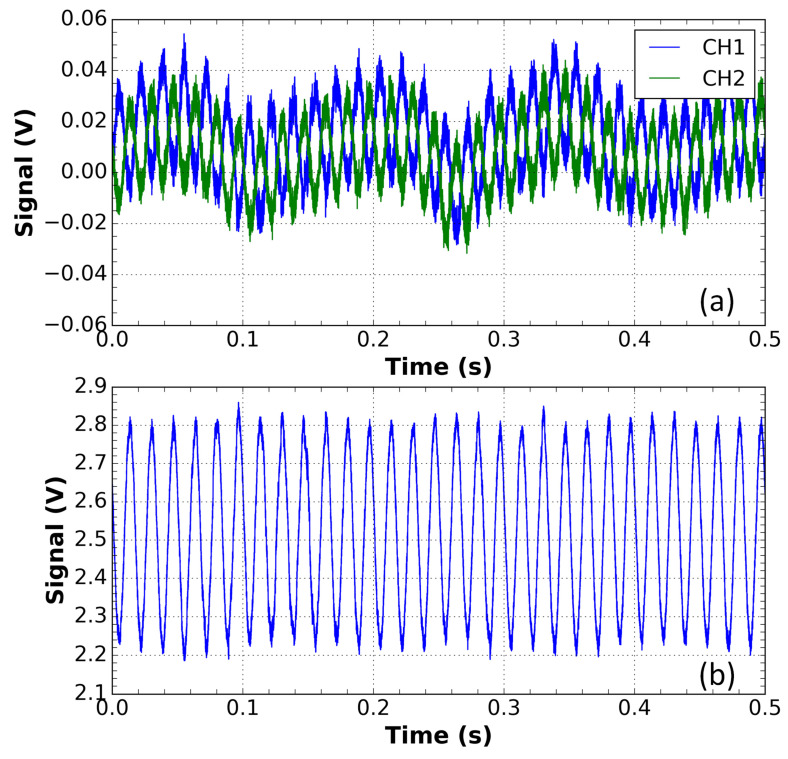
Measurements for the sensor inside the DWDM network. (**a**) Waveforms of the CH1 and CH2 channels of the voltage sensor output in the presence of a disturbance with an applied voltage of 9.56 kV. These signals were subtracted to compensate for the amplitude variation; (**b**) the result of this subtraction.

**Figure 12 sensors-21-04528-f012:**
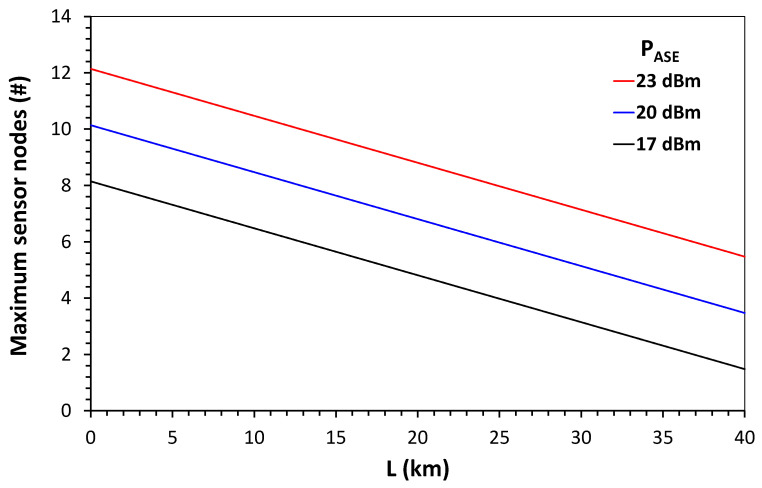
Plot of the number of nodes versus the link length (L) using (1) and the data cited above for three levels of ASE source optical power P_ASE_.

**Table 1 sensors-21-04528-t001:** Details of the light sources used in the first experiment. The least-square error (LSM) and the sum of the error bar were evaluated as the criterion of choice.

Source Name	Source Type	Power (dBm)	LSM	$\sum \sigma_{i}$
A	Compact ASE Source	8	1.54	1.27
B	High Power EDFA—Operation Type 1	4	4.87	1.55
C	High Power EDFA—Operation Type 2	20	11.70	12.58
D	High Power EDFA—Operation Type 3	20	2.47	3.47
E	SLED Source	11	1.85	0.25

## Data Availability

Data are contained within the article.

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
