# Peer review of "Study of a Current and Voltage Polarization Sensor Network"

_sensors, 2021, doi:10.3390/s21134528_

Round 1

Reviewer 1 Report

In this manuscript, the authors present an experimental study of a current and voltage polarization sensors network aims to be used in the medium voltage. The current and voltage polarization sensors are based on Faraday effect and Pockels effect, respectively, which is demonstrated in their previous work in the reference 16. The authors study the performance of these sensor systematically and comprehensively, and prove that these sensors have good stability and practicability.

However, the authors claimed that this is the first demonstration of an optical network using polarization sensors. I cannot agree with this. The system demonstrated in the manuscript is a DWDM system based on ADD/DROP. As presented in the manuscript, there are 6 ADD/DROPs in the sensor node. That worries me a lot about the cost of the system. The maximum nodes quantity is possible to reach 10 nodes of current and voltage sensors in a 10 km optical link according to the authors’ estimation. Compared with the traditional electrical networking scheme, the advantages of the proposed scheme should be evaluated objectively.

Additionally, there are several typographical errors in the manuscript. For example, it seems to be “important” but not “import” in line 36.

Reviewer 2 Report

A very good article with a correct structure, not too extensive but sufficient literature review and large, clear figures. I found no editorial errors.

Reviewer 3 Report

The authors present an experimental study on polarimetric current and voltage sensors operating with an DWDM network. The study will be interesting to readers concerned with instrumentation for power networks. Although polarimetric sensors are not new and have been explored in great detail over the past 4 decades, their combination with the DWDM network has some merits.

Unfortunately, the description of the sensing elements is extremely superficial. So is the description of the sensor node. In addition to the photos of the sensors, it is mandatory that the authors provide a detailed drawing of the optical set up. Similarly, for the sensor node, the authors are required to provide the details of the add/drop devices and how these are integrated with the sensor elements. Details of the interleaver are also missing. Furthermore, it is unclear how variable optical loss compensation is achieved. The description of the compensation method is extremely superficial, pointing out to 'various impairments on the signal' and merely stating that a difference-over-sum is applied but no physical reason for this is given. What impairments are the authors referring to? What are the physical reasons for these impairments?

The entire manuscript requires an overhaul in terms of English language. To help the authors to make the technical and language corrections I have appended the manuscript with highlights of some problematic text.
